# A broadband and strong visible-light-absorbing photosensitizer boosts hydrogen evolution

Ping Wang[1], Song Guo[1], Hong-Juan Wang[1], Kai-Kai Chen[1], Nan Zhang[1], Zhi-Ming Zhang [1] & Tong-Bu Lu[1,2]

Developing broadband and strong visible-light-absorbing photosensitizer is highly desired for dramatically improving the utilization of solar energy and boosting artificial photosynthesis. Herein, we develop a facile strategy to co-sensitize Ir-complex with Coumarins and boron dipyrromethene to explore photosensitizer with a broadband covering ca. 50% visible light region (Ir-4). This type of photosensitizer is firstly introduced into water splitting system, exhibiting significantly enhanced performance with over 21 times higher than that of typical Ir(ppy)$_2$(bpy)$^+$, and the turnover number towards Ir-4 reaches to 115840, representing the most active sensitizer among reported molecular photocatalytic systems. Experimental and theoretical investigations reveal that the Ir-mediation not only achieves a long-lived boron dipyrromethene-localized triplet state, but also makes an efficient excitation energy transfer from Coumarin to boron dipyrromethene to trigger the electron transfer. These findings provide an insight for developing broadband and strong visible-light-absorbing multi-component arrays on molecular level for efficient artificial photosynthesis.

---

[1] International Joint Research Laboratory of Materials Microstructure, Institute for New Energy Materials and Low Carbon Technologies, School of Materials Science & Engineering, Tianjin University of Technology, 300384 Tianjin, China. [2] MOE Key Laboratory of Bioinorganic and Synthetic Chemistry, School of Chemistry, Sun Yat-Sen University, 510275 Guangzhou, China. Correspondence and requests for materials should be addressed to S.G. (email: guosong@email.tjut.edu.cn) or to Z.-M.Z. (email: zmzhang@email.tjut.edu.cn) or to T.-B.L. (email: lutongbu@tjut.edu.cn)

Sunlight-driven water splitting into hydrogen has been regarded as an attractive strategy to simultaneously address the issues of energy crisis and environmental pollution[1–11]. As is well known, the solar spectrum exhibits a wide range from ultraviolet region (UV) to infrared region (IR), especially, the visible region covers >40% of solar spectrum[12]. As a result, developing broadband and strong visible-light-absorbing (BSVLA) photosensitizers (PSs) is highly desired for dramatically improving the utilization of solar energy[13–15]. In this field, major progress has been achieved in developing high-efficient catalysts and catalyst–PS coupling molecular devices for efficient hydrogen evolution[2,5,7,8,15–21]. However, the exploitation of highly active PSs are seriously delayed, although they play a critical role in absorbing solar light and mediating electron transfer among photocatalytic system[5,8,22–24].

In the past decades, noble metal Ru(II)-, Ir(III)-, Pt(II)-, and Re (I)-based molecular complexes, such as $Ru(bpy)_3^{2+}$, $Ir(ppy)_2(bpy)^+$, and $[Pt(tpy)(arylacetylide)]^+$ (tpy = terpyridine),[16,19–30] are frequently used as PSs for photocatalytic hydrogen evolution due to their long-lived triplet state ($^3$MLCT), which can supply enough time for electron transfer between different components[13,14,22,25–36]. However, the typical MLCT (metal-to-ligand charge transfer) PSs, such as $Ir(ppy)_2(bpy)^+$ (**Ir-1**), often suffer from weak visible light absorption ability ($\varepsilon < 15{,}000\,M^{-1}\,cm^{-1}$) and narrow visible light absorption range, which severely restricted their photocatalytic activity[13,14,37]. As is well known, organic dyes, such as boron dipyrromethene dye (Bodipy), Coumarin, and Fluorescein, generally exhibit high molar absorption coefficients ($\sim\!10^5\,M^{-1}\,cm^{-1}$) due to the existence of $\pi$–$\pi^\star$ transition on irradiation[13,14,38]. Nevertheless, most of these pure organic chromophores are inactive for photocatalytic hydrogen evolution because of their short-lived excited states (<10 ns)[13,31–34]. Against this background, implanting strongly absorbing chromophores into metal complexes has been regarded as an effective strategy to merge the advantages of metal complexes with $^3$MLCT state and organic dyes with $\pi$–$\pi^\star$ transition[15,37,39,40]. In this field, Bodipy, Rhodamine, and Coumarin were widely used to decorate $PtN_2S_2$ and Ir(III) complexes to construct strong visible-light-absorbing PSs[15,36,39]. However, these mono-chromophore-based PSs usually possess narrow visible light absorption range, resulting in low efficiency for solar energy utilization. Integrating two or more different chromophores into the same complex to construct BSVLA PSs for photocatalytic water splitting are still a great challenging task, as it requires efficient intersystem crossing (ISC) process to attain long-lived triplet state and efficient synergism between different chromophores, as well as suitable redox potentials to acquire supporting thermodynamic driving force. Up to date, several attempts have been devoted to exploring BSVLA PSs; however, few of them are competent to absorb solar light and mediate electron transfer in artificial photosynthetic systems to achieve highly efficient water splitting[41].

Herein a co-modification strategy by decorating Ir-based complex with different antennas is explored to construct BSVLA PSs. The resulting Coumarin and Bodipy co-decorating Ir-based PS (**Ir-4**) display strong visible-light-absorbing ability in the range of 400–575 nm, covering ca. 50% of visible light region. **Ir-4** was first used as a highly efficient PS for boosting hydrogen evolution, which exhibits outstanding performance under both 450 and 525 nm light-emitting diode (LED) irradiation. The turnover number (TON) of **Ir-4** can reach to 115,840 under optimized condition, which significantly outperforms those of other PSs, and is >320 times higher than that of **Ir-1**, demonstrating that **Ir-4** is the most active PS among all the reported molecular systems. It is noteworthy mentioning that the Coumarin antenna in **Ir-4** can efficiently convert excitation energy to Bodipy by the tandem processes of ISC and triplet–triplet energy transfer (TTET) via Ir mediation.

## Results

**Synthesis and characterization**. The principle for the design of **Ir-4** PS is to directly attach $\pi$ core of both Coumarin 6 and Bodipy to heavy atom center of Ir(III), which can efficiently mediate the excitation energy of antennas into long-lived triplet state and broaden the visible-light-harvesting range of PSs[14]. For comparison, **Ir-1**, **Ir-2**, and **Ir-3** were also synthesized (Supplementary Fig. 1). **Ir-1**–**Ir-4** were synthesized via a two-step reaction, with the yields >60% (Supplementary Methods)[37,40,42]. In a typical process, the ppy and Coumarin 6 were used to coordinate with Ir(III) to obtain Ir(III) intermediates of $[Ir(ppy)_2Cl]_2$ and $[Ir(Coumarin\ 6)_2Cl]_2$, respectively. Then the $[Ir(ppy)_2Cl]_2$ dimer reacts with bpy and bpy$-\equiv-$Bodipy to produce **Ir-1** and **Ir-3**, respectively. **Ir-2** and **Ir-4** were prepared by a similar method with that of **Ir-3** except using $[Ir(Coumarin\ 6)_2Cl]_2$ to replace $[Ir(ppy)_2Cl]_2$ (Supplementary Fig. 1). For **Ir-4**, the cyclometalated complexing and $\pi$-conjugation effect made both Coumarin 6 and Bodipy units closely surround Ir(III) center to maximize the heavy atom effect. All these complexes were well defined and characterized (Fig. 1, Supplementary Figs. 2–15), and the structure of **Ir-4** was further confirmed by $^{13}$C nuclear magnetic resonance (NMR) (Supplementary Fig. 16).

**Photocatalytic hydrogen evolution**. To enable better utilization of solar light, exploratory research using BSVLA complex as PS was explored by introducing Coumarin-Ir-Bodipy (**Ir-4**) into hydrogen evolution system (Fig. 2, Supplementary Figs. 17–21, and Supplementary Tables 1 and 2). Photocatalytic activity of **Ir-1**–**Ir-4**-containing systems was evaluated in $CH_3CN/H_2O$ (v/v = 9/1) with $[Co^{III}(dmgH)_2(py)Cl]$ (**C-1**) as catalyst and **DMT** as sacrificial electron donor (Supplementary Fig. 2). As shown in Fig. 2a, ca. 32 μmol $H_2$ was evaluated from **Ir-4**-containing system in 12 h, which was significantly enhanced compared to that of **Ir-1**, >21 times higher than that of **Ir-1**-containing system. Notably, TON of **Ir-4** can reach as high as 115,840 under optimized condition, which represents the most efficient sensitizer among all the reported molecular systems (Supplementary Fig. 19)[16,43]. Under this condition, the photocatalytic activities of **Ir-1**–**Ir-3**-containing systems were also investigated, showing the TONs of 361, 22,560, and 8270, respectively. The TON of **Ir-4** is >320 times higher than that of **Ir-1**, demonstrating that **Ir-4** is indeed a state-of-the-art PS. Furthermore, when the concentration of **Ir-4** was increased from 1.25 to 30 μM, the yield of hydrogen evolution increased from 32.1 to 120.7 μmol under the irradiation of 175 W Xenon with a 420-nm filter (Fig. 2b).

Importantly, photocatalytic activity of **Ir-4** is almost the sum of **Ir-2** and **Ir-3**, indicating efficient transfer of excitation energy from Coumarin to Bodipy in **Ir-4** (Supplementary Table 1). Further, isolated Coumarin 6 and Bodipy ligands were simultaneously introduced into the photocatalytic system of **Ir-1** with the ratio of Coumarin 6, Bodipy, and **Ir-1** same to that in the formula of **Ir-4**, and photocatalytic study reveals that the activity of the mixture is similar to that of **Ir-1**, far lower than that of **Ir-4**-containing system (Supplementary Table 2, Entry 9). These results indicate that Coumarin 6 and Bodipy cannot sensitize **Ir-1** due to the far distance between the antenna molecules and **Ir-1** in the physical mixture. Also, in the absence of **Ir-1**, trace amount of $H_2$ was produced with Coumarin 6 and Bodipy ligands as the PSs, revealing that the Ir mediation can enable inactive pigments as efficient light-harvesting molecules for water splitting. In addition, no or little hydrogen was detected in the absence of light, **C-1**, **DMT**, PS, and $H_2O$, manifesting that all above factors were indispensable for efficient hydrogen evolution (Supplementary Table 2). All the above results indicate that the BSVLA ability

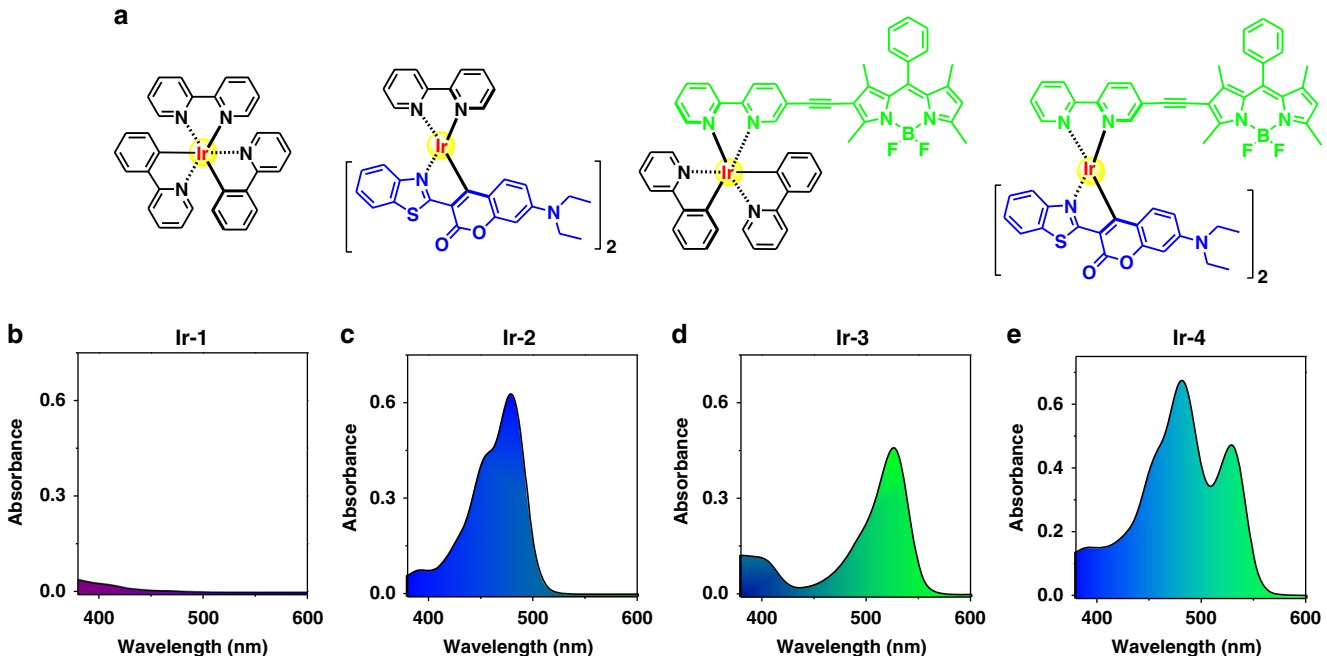

**Fig. 1** Molecular structures and absorption spectra of photosensitizers. **a** Molecular structures and ultraviolet−visible absorption spectra of **b Ir-1**, **c Ir-2**, **d Ir-3**, and **e Ir-4** in $CH_3CN$ (5 μM)

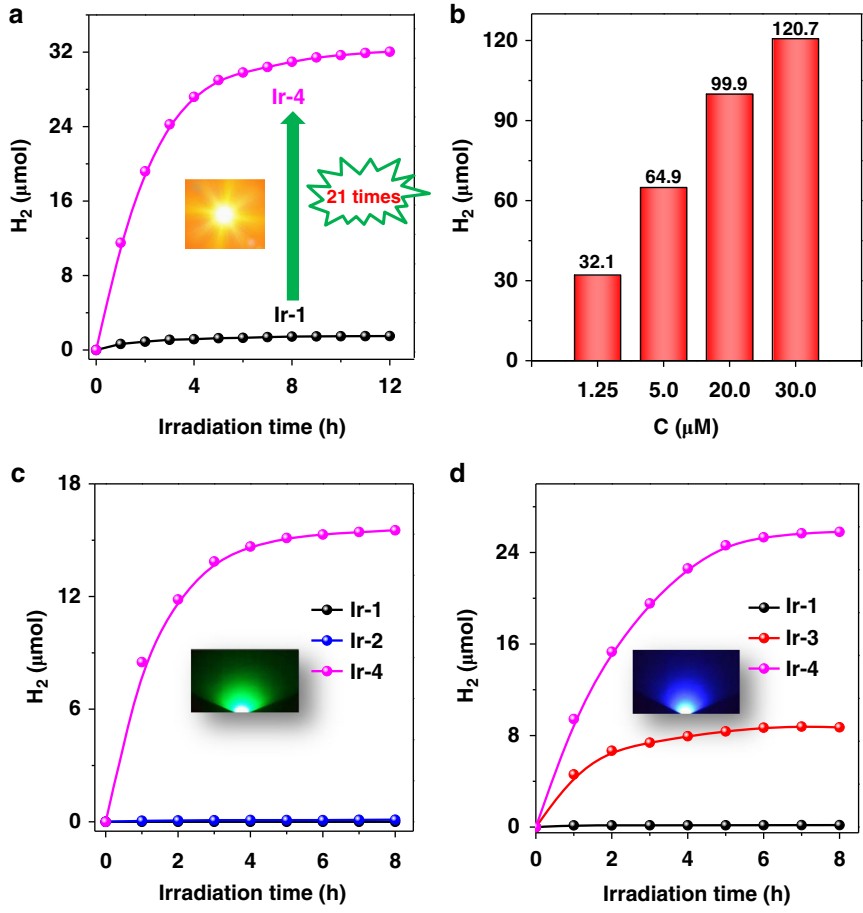

**Fig. 2** Photocatalytic hydrogen evolution. Photocatalytic hydrogen evolution with irradiation of **a** Xe lamp ($\lambda > 420$ nm, 175 W), **b** Xe lamp ($\lambda > 420$ nm, 175 W) for different concentrations of **Ir-4**, **c** 525 nm and **d** 450 nm light-emitting diode with light intensity of 100 mW cm$^{-2}$. Conditions: catalyst (50.0 μM), PS (1.25 μM) and **DMT** (0.01 M) in $CH_3CN/H_2O$ (v/v = 9/1)

of **Ir-4**, resulting from closely decorating Ir(III) center with Coumarin 6 and Bodipy, greatly contributes to boosting its photocatalytic activity.

In order to further highlight the advantage of BSVLA **Ir-4**, photocatalytic experiments were carried out with 450 and 525 nm LED (corresponding to the absorption of Coumarin 6 and Bodipy, respectively) as monochromatic light source, respectively (Fig. 2c, d). The photocatalytic activity of **Ir-4** was obviously superior to that of **Ir-3** and **Ir-2** upon excitation at 450 and 525 nm, respectively, indicating that **Ir-4** was potential for breaking the limitation of narrow absorption band of mono-chromophore PSs. **Ir-1** gives an extremely weak activity in hydrogen evolution upon LED irradiation at either 450 or 525 nm due to its poor visible-light-harvesting ability[42]. In addition, the apparent quantum yield of **Ir-4** was estimated to be 37.7% at 475 nm and 25.1% at 520 nm based on the incident photons[44,45]. In order to ascertain the actual source of protons, the photocatalytic products in the presence of $H_2O$ and $D_2O$ were both studied by using mass spectrometry (MS) analysis (Supplementary Fig. 21)[46,47]. $H_2$ as the sole gaseous product was detected by MS in the mixed solvents of $H_2O$ and $CH_3CN$, and $D_2$ became the major product when replacing $H_2O$ with $D_2O$. In addition, only trace amount of hydrogen can be detected with pure $CH_3CN$ as solvent in the absence of $H_2O$. These results confirm the $H_2O$ as the source of protons and exclude the dehydrogenation of **DMT** or $CH_3CN$. Systematic investigation of steady spectra, transient spectra, electrochemistry, and density functional theory (DFT) calculation reveal that the improved activity of **Ir-4** can be ascribed to the comprehensive factors, such as strong visible-light-harvesting ability, efficient electron transfer, suitable redox potential, and excited state type with long excited state lifetime.

**Steady absorption and emission spectra.** To decipher the intrinsic properties of PSs and electron transfer process between different components, steady spectra of PS alone and PS with **C-1** or **DMT** were performed, respectively. UV-vis absorption spectra of **Ir-2** and **Ir-3** show strong absorption band around 475 and 530 nm, corresponding to the absorption of Coumarin 6 and Bodipy, respectively (Supplementary Fig. 22)[37,40,42]. **Ir-4** shows a broad absorption band between 400 and 575 nm, which is almost superposition of the absorption of **Ir-2** and **Ir-3** (Supplementary Fig. 22a). This result indicates no electronic interaction between Bodipy and Coumarin 6 in **Ir-4** under the ground state. Furthermore, UV-vis absorption spectra of PS alone, PS with **C-1**, PS with **DMT**, and PS with both **C-1** and **DMT** were all studied in detail (Supplementary Fig. 23). Related results show that the absorption spectra of **Ir-2**–**Ir-4** stay almost unchanged before and after adding **DMT**, **C-1**, or **DMT** and **C-1**, indicating no electronic interaction between PSs in the ground state and **DMT** (or **C-1**).

The emission spectra of **Ir-1**–**Ir-4** were studied under different atmospheres (Fig. 3a–d and Supplementary Figs. 24–25). In nitrogen, **Ir-1** gave an emission peak around 585 nm, and **Ir-2**–**Ir-4** show dual emission peaks at 507/586, 556/745, and 506/557 nm, respectively. The peaks at 585 nm for **Ir-1**, 586 nm for **Ir-2**, and 745 nm for **Ir-3** became weaker or disappeared in the air, indicating that these peaks could be attributed to phosphorescence (PL) originating from triplet excited states (TESs) of these complexes[13,40,42]. No significant change for other peaks under different atmospheres manifested that these peaks should be the residual fluorescence (FL) of chromophores, derived from their singlet excited states (SESs). These results were further verified by emission lifetime (Fig. 3e and Supplementary Fig. 26). No or weak PL was observed for **Ir-4** and **Ir-3** due to the presence of Bodipy-localized $^3IL$ state[42,48]. Isolated Coumarin 6 and Bodipy usually showed strong FL; however, their FL was completely quenched in

**Ir-2**–**Ir-4** owing to an efficient ISC process from SES to TES (Supplementary Fig. 24).

In this process, the emission quenching experiments of Ir(III) complexes by **DMT** and **C-1** were carried out to study the electron transfer efficiency (Fig. 3b, c and Supplementary Figs. 27–29). The efficiency of FL quenching of **Ir-2** and **Ir-3** by **DMT** and **C-1** is far lower than that of PL quenching, indicating that long-lived triplet state is more beneficial to electron transfer than short-lived singlet state[22,38]. Thus the initial step of electron transfer process should be dominated by triplet states of Ir(III) complexes. The efficiency of electron transfer can be determined by PL quenching experiments. The $K_{sv}$ quenching constants of PSs quenched by **DMT** were in the order of **Ir-3** ($407,200 \, M^{-1}$) > **Ir-1** ($3607 \, M^{-1}$) > **Ir-2** ($224 \, M^{-1}$), proportional to their triplet lifetimes (Supplementary Table 3). As shown in Fig. 3f, upon 510 nm excitation, the SES of **Ir-3** was populated and then afforded a triplet state via ISC process. As result, **Ir-3** can emit FL and PL simultaneously, providing the first molecular platform that can directly evaluate electron transfer ability between singlet state and triplet state in same one molecule with dual emission. For **DMT** quenching, the quenching constant of PL can reach to ca. $4.1 \times 10^5 \, M^{-1}$, >7000 times higher than that of the FL quenching ($56.2 \, M^{-1}$). A series of experiments reveal that the electron transfer efficiency from long-lived triplet state significantly outperforms that from short-lived excited state.

**Electrochemical study.** In order to assess the thermodynamic feasibility of electron transfer, redox potentials of **Ir-1**–**Ir-4**, **DMT**, and **C-1** were determined in $CH_3CN/H_2O$ (v/v = 9/1) (Fig. 4, Supplementary Figs. 30–31, Supplementary Tables 1 and 4) by the cyclic voltammetry (CV) (Supplementary Methods). **Ir-1** gave an oxidation potential at 1.31 V and a reduction potential at −1.37 V, corresponding to the redox processes from $Ir^{3+}$ to $Ir^{4+}$ and $bpy^0$ to $bpy^{-1}$, respectively. For **Ir-2** containing two Coumarin 6 units, three reduction potentials at −1.19, −1.45, and −1.66 V can be attributed to Coumarin $6^{0/-1}$, Coumarin $6^{-1/-2}$, and $bpy^{0/-1}$, respectively. The first reduction potential of **Ir-3** was determined as −0.94 V, less negative than that of **Ir-1** and **Ir-2** as the existence of reduction process from Bodipy$^0$ to Bodipy$^{-1}$. **Ir-4** also shows three reduction potentials at −0.94, −1.12, and −1.46 V, respectively. The first reduction potential of **Ir-4** was same as that of **Ir-3**, as well as the second and third reduction potentials of **Ir-4** were very close to the first two reduction potentials of **Ir-2**, indicating that the reduction potentials of **Ir-4** correspond to the redox process of Coumarin 6 and Bodipy units, respectively (Table 1). Accordingly, there was hardly any electron interaction between Coumarin and Bodipy ligands in **Ir-4** under the ground state, which well matched the results of absorption spectra. Additionally, the oxidation potential of **Ir-2** was determined as 1.12 V, less positive than that of **Ir-1** and **Ir-3**, and **Ir-3** possesses of an oxidation potential of 1.28 V, very close to that of **Ir-1**. These results manifested that Coumarin had a more significant effect on the oxidation potential of Ir-based complexes in comparison with the Bodipy unit. As a result, it is reasonable that the oxidation potential of **Ir-4** (1.19 V) was close to that of **Ir-2**.

CV of **C-1** was also performed in deaerated $CH_3CN$ and $CH_3CN/H_2O$ (v/v, 9/1) solution to uncover its catalytic mechanism (Supplementary Fig. 32)[49–51]. The redox process of $Co^{2+}/Co^+$ at −1.09 V (vs. saturated calomel electrode (SCE)) remained reversible in water-containing system, indicating that it is impossible to carry out catalytic water splitting at this potential[49]. Interestingly, a large catalytic current emerged at around −1.48 V (vs. SCE) corresponding to the redox process of $Co^+/Co^0$, which could be attributed to the proton reduction process in the presence of Co(0) species[51]. Considering that the

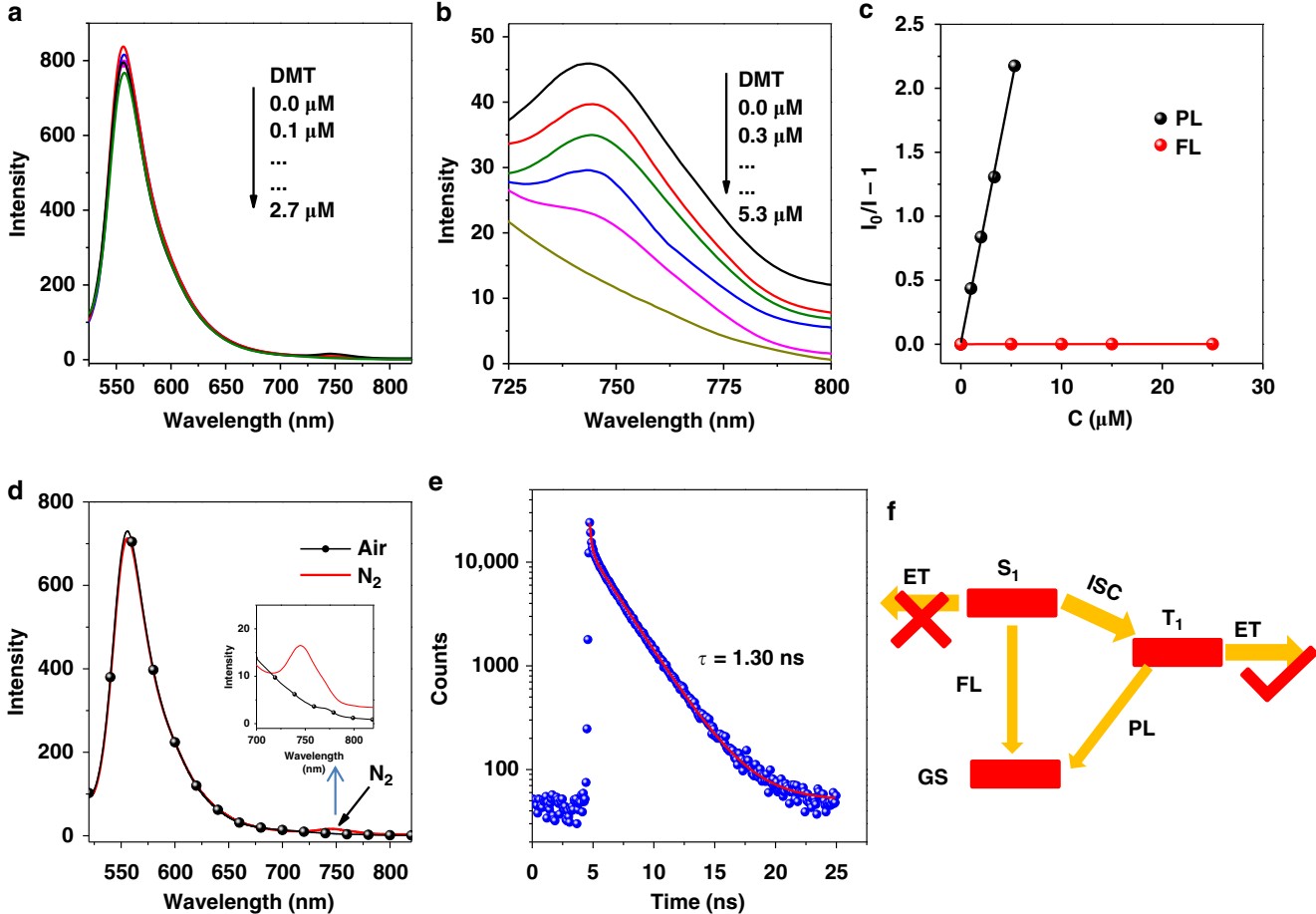

**Fig. 3** Emission quenching. **a** Ir-3 with **DMT** ($\lambda_{ex} = 510$ nm), **b** zoom in the range between 725 and 800 nm, **c** Stern–Volmer plot of fluorescence and phosphorescence by **DMT**; **d** the emission of **Ir-3** under nitrogen and air atmosphere; **e** emission lifetime of **Ir-3** at 556 nm, $\lambda_{ex} = 510$ nm, and **f** photophysical process of **Ir-3** upon light excitation. ET electron transfer, GS ground state, ISC intersystem crossing. $c_{PS} = 5.0$ μM

possibility of electron transfer from reduced PSs to Co$^+$ was thermodynamically ruled out, we tentatively proposed that the disproportionation of Co$^+$ into Co$^{2+}$ and Co$^0$ was a possible pathway in this photocatalytic system[52]. Further, electrolysis experiment of **C-1**-containing system was performed to confirm this view, where H$_2$ was indeed detected at −1.09 V (vs. SCE, corresponding to Co(II)/Co(I)), indicating the formation of Co$^0$ species during this electrolysis process (Supplementary Table 5).

The Gibbs free energy changes ($\Delta G_{CS}$) of electron transfer from **DMT** to excited PSs, from excited PS to **C-1**, and from reduced PSs to **C-1** were figured out by Weller equation (Supplementary Eqs. 1–3)[53–55]. The negative values of $\Delta G_{CS}$ for all the reduction processes indicate the thermodynamic feasible for relevant electron transfer. For the oxidation process, the electron transfer from excited **Ir-3** to Co$^{2+}$ and excited **Ir-4** to Co$^{2+}$ were restricted owing to the positive values of $\Delta G_{CS}$ (0.40 for **Ir-3** and 0.31 for **Ir-4**), which thermodynamically ruled out the possibility of oxidation mechanism. Therefore, photocatalytic route of these photocatalytic systems can be determined as the reduction mechanism according to the electrochemical results. The absolute values of $\Delta G_{CS}$ of excited **Ir-1/DMT** (0.44) and **Ir-2/DMT** (0.52), as well as reduced **Ir-1**/Co$^{2+}$ (0.45) and reduced **Ir-2**/Co$^{2+}$ (0.27), were much larger than those of related processes of **Ir-3** and **Ir-4**. These results demonstrated much larger driven forces from **DMT** to excited **Ir-1** and from reduced **Ir-1** to **C-1** in comparison with those of **Ir-3**- and **Ir-4**-containing systems. However, **Ir-1** exhibits the lowest activity among these four compounds, and this can be largely attributed to its poor visible-

light-harvesting ability. Further, **Ir-4** containing three additional antennas exhibits the best photocatalytic performance among **Ir-1**–**Ir-4**, although it shows a humble driven force for electron transfer. Hence, it can be proposed that the BSVLA ability and long-lived excited state of **Ir-4** plays an important role in dramatically improving its photocatalytic performance. As a result, **Ir-4** was determined to be the most efficient PSs among **Ir-1**–**Ir-4** for hydrogen evolution with the TON reaching 115,840, representing the most active sensitizer among all the reported molecular photocatalytic systems.

**Transient absorption spectra.** Nanosecond transient absorption spectra were investigated in degassing CH$_3$CN to shed light on the photocatalytic process (Figs. 5 and 6 and Supplementary Figs. 33–35). Upon excitation at 355 nm, **Ir-1** showed two positive absorption bands around 380 and 490 nm, similar to previous report, and its lifetime was determined as 307 ns (Supplementary Fig. 33)[42]. By addition of **DMT**, a long-lived species of reduced **Ir-1** was obtained with a decay of 63 μs (Supplementary Fig. 33e). In the presence of **C-1**, the decay of reduced **Ir-1** was significantly faster, supporting an extra electron transfer pathway from reduced **Ir-1** to **C-1**. For **Ir-2**, a strong bleaching peak around 475 nm was observed upon pulsed laser excitation, corresponding to its ground state absorption (Fig. 5a). The triplet state of **Ir-2** was localized on Coumarin 6 with a triplet state lifetime of 85 ns (Fig. 5d). After adding **DMT**, the bleaching peak shows a redshift from 475 to 487 nm, and two new positive absorption bands

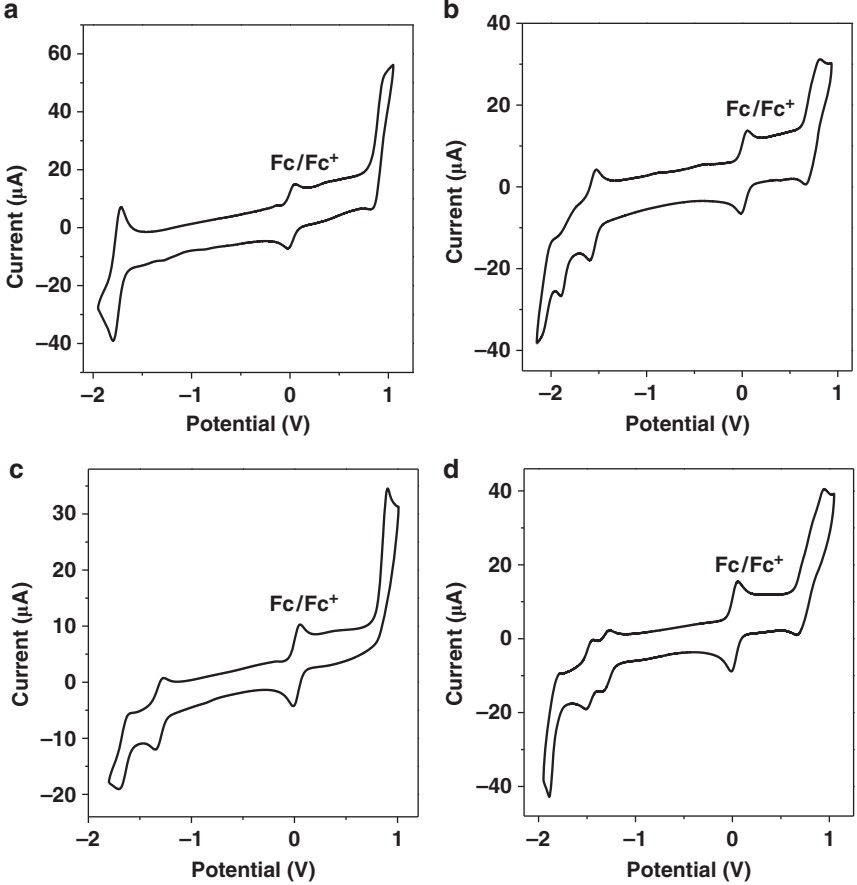

**Fig. 4** Electrochemical study. Cyclic voltammograms of **a Ir-1**, **b Ir-2**, **c Ir-3**, and **d Ir-4** were determined in deaerated $CH_3CN/H_2O$ (v/v, 9/1) solution, containing 0.5 mM photosensitizer, ferrocene, and 0.10 M $Bu_4NPF_6$ as the supporting electrolyte, with a scan rate of 0.05 V/s$^{-1}$ and a negative initial scan direction. Glassy carbon electrode, Ag/AgNO$_3$, and Pt silk were used as the working electrode, reference electrode, and counter electrode, respectively

**Table 1 Redox potentials of Ir-1–Ir-4 and the $\Delta G_{CS}$ for intermolecular electron transfer**

|        | $E_{ox}$ | $E_{red}$            | $E_{0,0}$ | $\Delta G_{CS}^{a}$ | $\Delta G_{CS}^{b}$ | $\Delta G_{CS}^{c}$ | $\Delta G_{CS}^{d}$ | $\Delta G_{CS}^{e}$ |
|--------|----------|---------------------|-----------|---------|---------|---------|---------|---------|
| **Ir-1** | 1.31 | −1.37              | 2.53 | −0.44 | −1.09 | −0.30 | −1.24 | −0.45 |
| **Ir-2** | 1.12 | −1.19, −1.45, −1.66 | 2.43 | −0.52 | −1.18 | −0.39 | −1.03 | −0.27 |
| **Ir-3** | 1.28 | −0.94, −1.28        | 1.80 | −0.14 | −0.39 | +0.40 | −0.81 | −0.02 |
| **Ir-4** | 1.19 | −0.94, −1.12, −1.46 | 1.80 | −0.14 | −0.48 | +0.31 | −0.81 | −0.02 |
| **DMT**  | 0.76 | —                  | —    | —     | —     | —     | —     | —     |
| **C-1**  | —    | −0.13, −0.92        | —    | —     | —     | —     | —     | —     |

The potential values are given with respect to saturated calomel electrode (SCE) ($F_c$ as internal reference, $E_{1/2(Fc+/Fc)} = + 0.40$ V vs. SCE)
$\Delta G_{CS}$ represents the value of the change in Gibbs free energy for the electron transfer process: $^{a}$from **DMT** to excited PS, $^{b}$from excited PS to Co$^{3+}$ (**C-1**), $^{c}$from excited PS to Co$^{2+}$ (**C-1**), $^{d}$from reduced PS to Co$^{3+}$ (**C-1**), $^{e}$from reduced PS to Co$^{2+}$ (**C-1**)

between 380/460 and 525 nm/550 nm appeared, indicating the formation of new transient species with a long-lived decay at 500 nm (25.0 μs) (Fig. 5b, e). This new species can be identified as the reduced state of **Ir-2**, further supported by PL quenching experiments. The reduced **Ir-2** with **C-1** shows a faster decay to baseline than that of reduced **Ir-2** alone, manifesting an efficient electron transfer from reduced **Ir-2** to **C-1** (Fig. 5c, f). As a result, the photocatalytic process of **Ir-1**- and **Ir-2**-containing systems is mainly dominated by reduction mechanism.

Compared to **Ir-2**, transient absorption spectrum of **Ir-3** exhibits a strong bleaching peak at 525 nm upon pulsed laser excitation at 532 nm, which well matched with the ground state absorption of Bodipy (Supplementary Fig. 34)[42]. Hence, triplet state of **Ir-3** was populated on the Bodipy part, further confirmed

by its long-lived excited state ($\tau_T = 101$ μs) (Supplementary Fig. 34a, d). In the presence of **DMT**, both the positive peak around 430 nm and the bleaching peak at 523 nm obviously decreased, meanwhile, a new peak at 580 nm rose, indicating the formation of a new species (Supplementary Fig. 34b). The lifetime of this species was up to 113 μs, even longer than that of **Ir-3** alone (Supplementary Fig. 34e). As is well known, the triplet state of **Ir-3** could be efficiently quenched by **DMT** through an electron transfer process. Accordingly, we proposed the new species as the reduced **Ir-3**. Similar to **Ir-2**-containing system, transient spectra confirmed that the reduced **Ir-3** can be efficiently oxidized by **C-1** (Supplementary Fig. 34c, f). As a result, reduction mechanism for **Ir-3**-containing system should be a dominant electron transfer process.

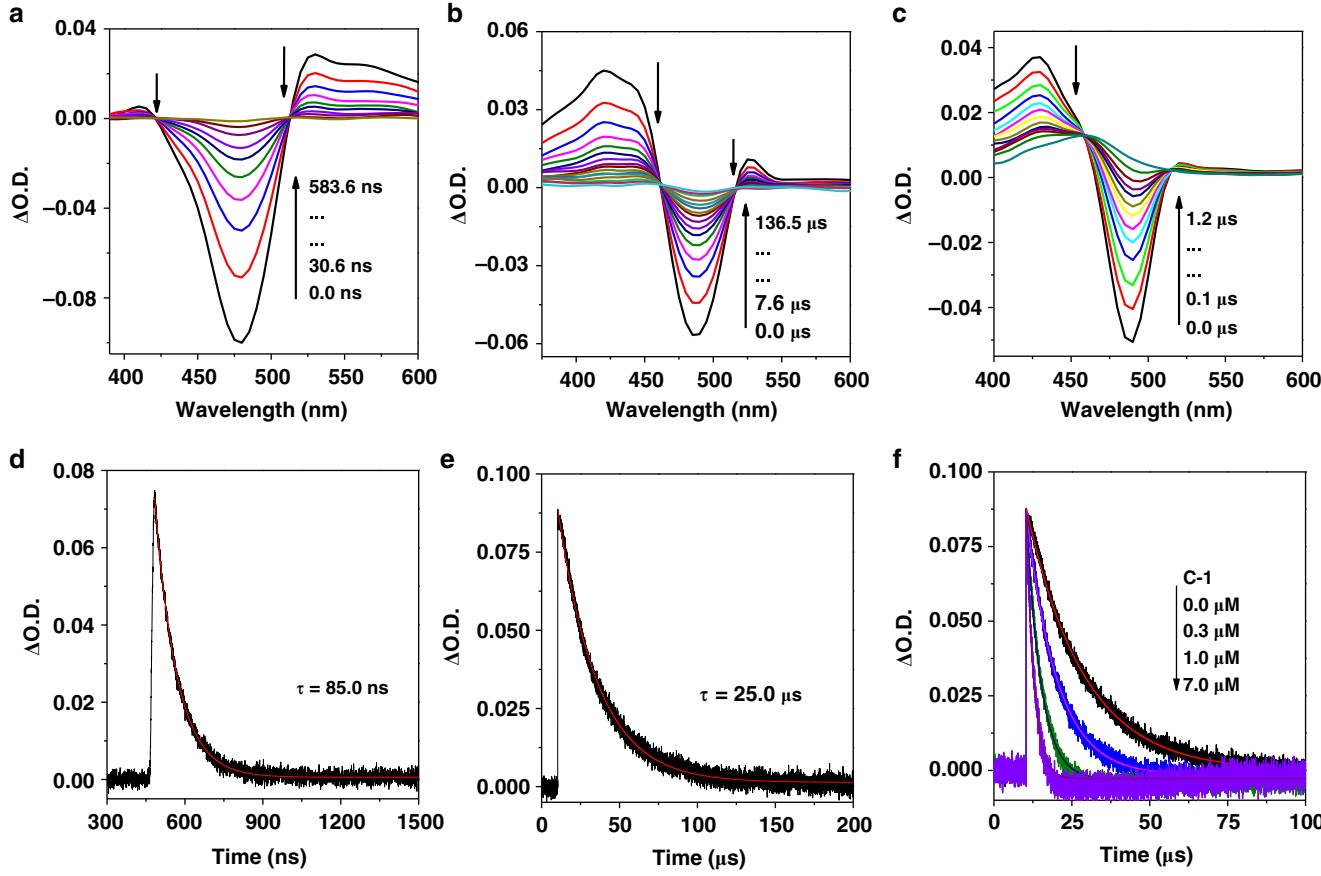

**Fig. 5** Nanosecond transient absorption spectra of Ir-2. **a** Ir-2, **b** Ir-2 with 40 mM of **DMT**, **c** reduced **Ir-2** with 0.2 mM of **C-1**; **d** the decay of **Ir-2** at 468 nm, **e** kinetic traces of reduced **Ir-2** followed at 500 nm, **f** kinetic traces of reduced **Ir-2** with different concentration of **C-1** followed at 500 nm. These spectra were recorded in CH₃CN after pulsed excitation at 355 nm under N₂

The photophysical processes of **Ir-4** with multiple chromophores should be more complex in comparison with that of **Ir-2** or **Ir-3** (Fig. 6). Upon excitation at either 532 or 470 nm, the transient spectrum of **Ir-4** was similar to that of **Ir-3**, indicating that the excited state of **Ir-4** was also distributed on the Bodipy part (Fig. 6a, d). This can be ascribed to the lower triplet energy level of Bodipy than that of Coumarin antenna. Notably, transient spectra of **Ir-4** remained unchanged upon exciting Coumarin and Bodipy ligands, signifying that the Coumarin antenna could deliver fast excitation energy to Bodipy ($>10^8 \, \text{s}^{-1}$) (Supplementary Fig. 35). After adding **DMT**, a long-lived reduced state of **Ir-4** (88.7 μs) was obtained, verified by the similar method of **Ir-3** (Fig. 6b, e). In the presence of **C-1**, the transient spectra of reduced **Ir-4** returned more quickly to the baseline, and the decay at 525 nm became more faster by increasing the concentration of **C-1** with a second-order rate constant of $8.1 \times 10^{10} \, \text{M}^{-1} \, \text{s}^{-1}$, revealing an efficient electron transfer from reduced **Ir-4** to **C-1** (Fig. 6c, f). Consequently, the reduction quenching pathway should be a dominant process for **Ir-4**-containing system. To further confirm the presence of reduced species of the PSs, spectroelectrochemical (SEC) experiments were performed in the Ar atmosphere[43]. As shown in Supplementary Fig. 36, the transient absorption spectra of **Ir-1**–**Ir-4** with **DMT** matched well with the absorption of reduced PSs but were different from that of the oxidized ones (Supplementary Fig. 37). Thus the transient absorption spectra of **Ir-1**–**Ir-4** with **DMT** could be assigned to the absorption of reduced PSs.

Femtosecond transient absorption spectroscopy of **Ir-4** was carried out to clarify its intramolecular energy transfer process

(Supplementary Fig. 38)[56–58]. Upon selective excitation at 438 nm (corresponding to Coumarin ligand), both the bleaching peak at 481 nm and excited state absorption band beyond 600 nm for the triplet state of Coumarin 6 decreased, accompanied by the enhancement of the bleach signal around 528 nm and excited state absorption band at 440 nm of the triplet state of Bodipy (Figs. 5a and 6a). Therefore, this process can be attributed to intramolecular TTET and its rate constant was determined to be $k_{\text{TTET}} = 4.3 \times 10^{10} \, \text{s}^{-1}$ by monitoring the increase of bleaching band at 528 nm. It should be noted that no ISC process can be observed with this femtosecond transient absorption spectroscopy, indicating an ultrafast ISC process. This can be ascribed to the fact that the cyclometalated complexing approach made Coumarin 6 closely surround Ir(III) center to maximize the heavy atom effect.

Both **Ir-3** and **Ir-4** show long-lived triplet state with the lifetime of 101 and 89 μs, respectively. By contrast, the triplet lifetimes of **Ir-1** and **Ir-2** were merely 307 and 85 ns, respectively, which is to the disadvantage of the electron transfer between different components. In addition, all these complexes show a long-lived reduced state (63.0, 25.0, 113.0, and 88.7 μs for **Ir-1**, **Ir-2**, **Ir-3**, and **Ir-4**, respectively), and the second-order rate constants from reduced **Ir-1**–**Ir-4** by **C-1** were determined as $2.0 \times 10^{10}$, $1.7 \times 10^{11}$, $2.8 \times 10^{10}$, and $8.1 \times 10^{10} \, \text{M}^{-1} \, \text{s}^{-1}$, respectively, which were proximate to the diffusion-controlled limits (Supplementary Fig. 39 and Supplementary Table 6)[59]. These results indicate that the long-lived triplet state for **Ir-3** and **Ir-4** can supply enough time for efficient electron transfer between PS and other components in these three component systems and

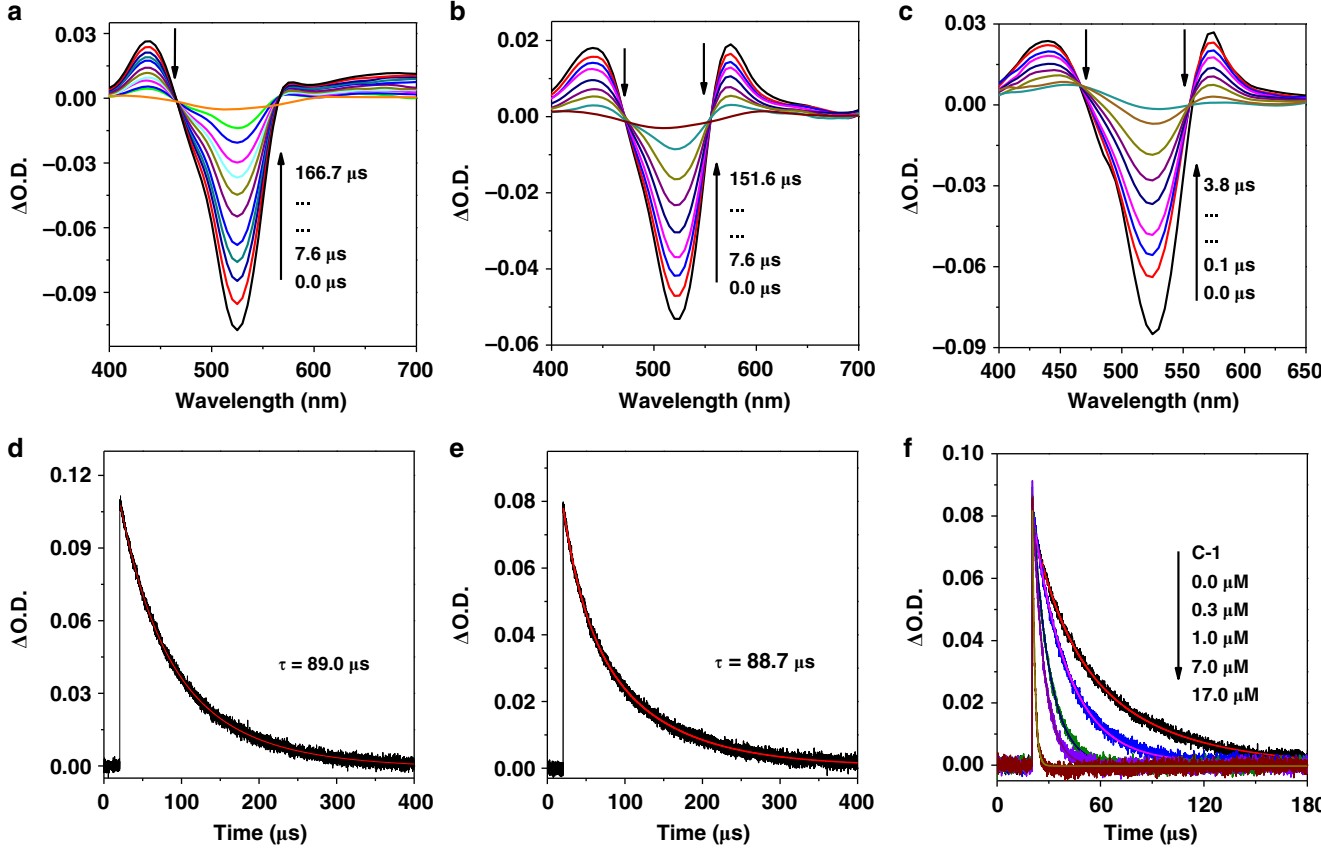

**Fig. 6** Nanosecond transient absorption spectra of Ir-4. **a** Ir-4, **b** Ir-4 in the presence of 40 mM of **DMT**, **c** reduced **Ir-4** in the presence of 0.2 mM of **C-1**, **d** the decay of **Ir-4** at 517 nm, **e** kinetic traces of the reduced **Ir-4** followed at 525 nm, **f** kinetic traces of reduced **Ir-4** with different concentration of **C-1** followed at 525 nm. These spectra were recorded in $CH_3CN$ after pulsed excitation at 532 nm under $N_2$

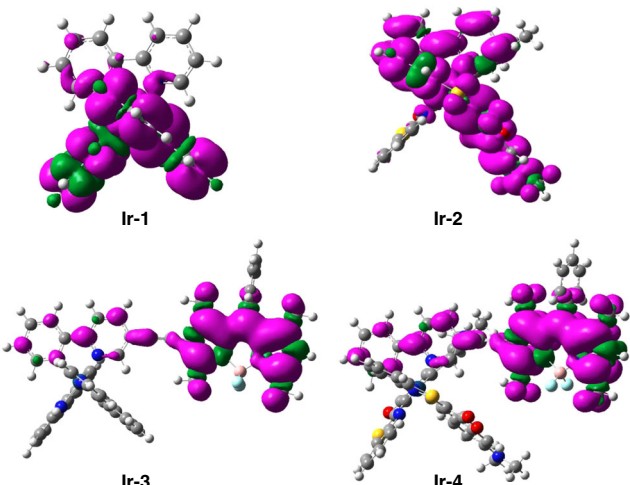

**Fig. 7** Spin density surfaces. Spin density surfaces of **Ir-1**–**Ir-4** were calculated at B3LYP/6–31G/genecp/LanL2DZ level with Gaussian 09

**Ir-4** possessed BSVLA ability, long-lived triplet state, and long-lived reduced state, highlighting that **Ir-4** could act as an efficient PS for promoting hydrogen evolution.

**DFT calculations on Ir-1–Ir-4.** DFT calculation represents a powerful method to unveil the property of excited states of PSs. Herein the population of excited states and photophysical processes of **Ir-4** were both evaluated by time-dependent DFT (TDDFT) method (Figs. 7 and 8). For **Ir-1**, the spin density was localized on ppy and Ir center, indicating the presence of both $^3$MLCT and $^3$IL states, which were also localized on bpy, Coumarin, and Ir center in **Ir-2**. The distribution of spin density can rationalize the short-lived TES of **Ir-1** and **Ir-2** (Fig. 5d). The spin density of both **Ir-3** and **Ir-4** mainly distributed on Bodipy part, the Ir(III) center, bpy, and Coumarin ligands made little contribution, indicating that their lowest-lying TESs could be attributed to Bodipy-localized $^3$IL state. Notably, these results are well consistent with the results of PL and transient absorption spectra.

In order to track the evolution of intramolecular photophysical processes of **Ir-4**, UV-vis absorption, FL, and TESs were evaluated based on optimized ground-state geometry with TDDFT method (Fig. 8 and Supplementary Table 7). For UV-vis absorption, two major transitions ($S_0 \rightarrow S_3$ and $S_0 \rightarrow S_{16}$), judged by the oscillator strength, were located at 589 and 418 nm and the corresponding molecular orbitals were distributed on Bodipy and Coumarin ligands in **Ir-4**, respectively. These results matched well with experimental results of absorption spectra. The calculated results reveal that the SESs of $S_1$ and $S_2$ were both the charge transfer states as their negligible oscillator strength. The transition of $S_3 \rightarrow S_0$ corresponded to conversion of electronic cloud distribution in Bodipy, which was in line with the emission spectra of **Ir-4**. As the heavy atom effect of Ir, the SES of **Ir-4** can efficiently transform into triplet state via an ISC process. The $T_1$ state was localized on the Bodipy unit, which fully agreed with transient absorption spectra and spin density surfaces of **Ir-4**. In addition, $T_5$ and $T_6$ states were populated on two Coumarin

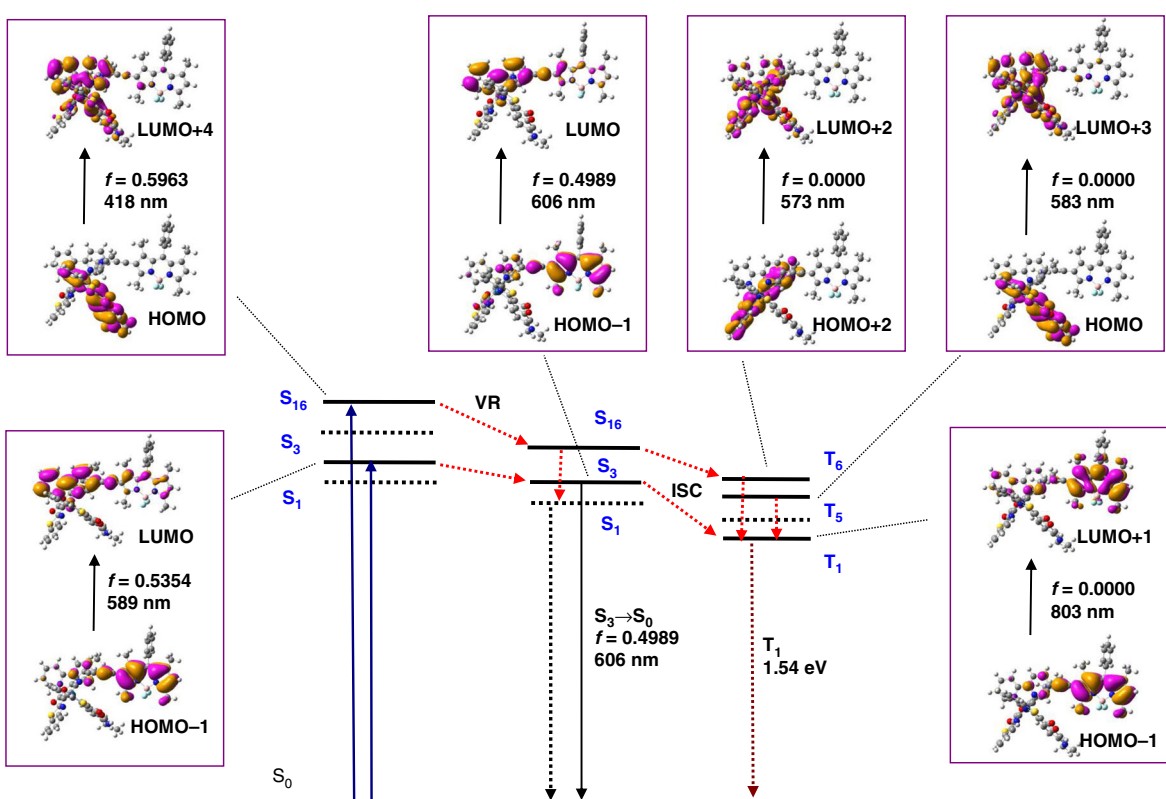

**Fig. 8** Selected frontier molecular orbitals involved in the excitation and singlet excited state/triplet excited state (TES) of Ir-4. VR stands for vibrational relaxation. The left column is ultraviolet–visible absorption (based on ground state geometry), the middle column is the fluorescence emission (based on $S_1$ state geometry), and the right column is the TES (based on ground state geometry). For clarity, only selected excited states are presented

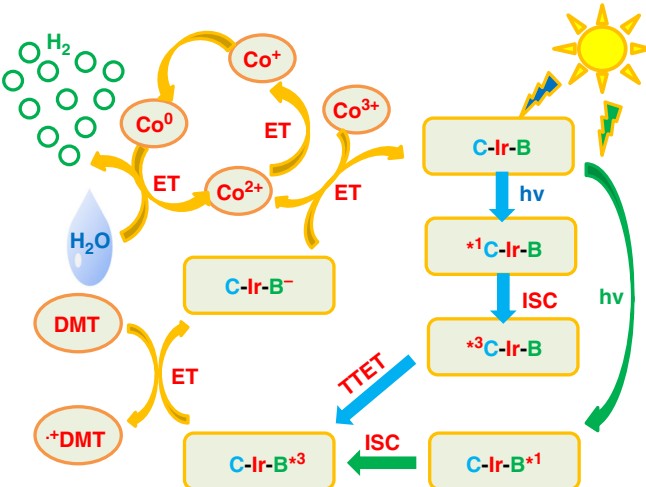

**Fig. 9** Proposed photochemical process for hydrogen evolution with Ir-4 (C-Ir-B). Ir is the coordination center, C is Coumarin, B is Bodipy, ET is electron transfer, ISC is intersystem crossing, TTET stands for triplet-triplet energy transfer

ligands and they were degenerate due to the similar energy level of triplet state (2.12 eV for $T_5$ and 2.16 eV for $T_6$). As a result, the whole photophysical processes of **Ir-4** were rationalized by DFT/TDDFT calculations (Fig. 8), confirming the efficient energy transfer from Coumarin to Bodipy and the population of the long-lived Bodipy-localized $^3IL$ state, which can efficiently trigger the electron transfer to afford redox reactions.

In a word, the photocatalytic cycle of **Ir-1**–**Ir-4**-containing systems proceeded via a reductive route. For **Ir-4**-containing system, both Bodipy and Coumarin ligands can be excited by visible light, and subsequently electron transfer proceeded by two different pathways. As shown in Fig. 9, upon exciting the Bodipy part, the triplet state of **Ir-4** was populated on Bodipy by a photochemical process of $[C-Ir-B] \rightarrow {}^1[C-Ir-B^*] \rightarrow {}^3[C-Ir-B^*]$. When exciting Coumarin unit, a more complex photophysical process was revealed as follows: $[C-Ir-B] \rightarrow {}^1[{}^*C-Ir-B^*] \rightarrow {}^3[{}^*C-Ir-B] \rightarrow {}^3[C-Ir-B^*]$. Hence, Bodipy-localized triplet state of **Ir-4** could accept electron from **DMT** to generate the reduced **Ir-4**, which further efficiently delivered electrons to **C-1** to produce Co(I). However, the possibility of electron transfer from reduced **Ir-4** to Co(I) was thermodynamically ruled out. Thus the formation of Co(0) species could be attributed to the disproportionation of Co(I) into Co(II) and Co(0)[52]. Finally, the protons were reduced to hydrogen by reduced **C-1**. As a result, **Ir-4** with multichromophores showed dual excitation channels, which can promote solar energy conversion and the subsequent photocatalytic water splitting.

## Discussion

In conclusion, Bodipy and Coumarin ligands with strong visible light absorption bands in different regions were delicately integrated into a Ir(III) complex to dramatically improve the utilization of solar energy. **Ir-4** exhibits a BSVLA range from 400 nm to 575 nm that can cover ca. 50% of visible light region. Systematic experimental investigations and DFT calculations thoroughly rationalize the photophysical process, where Ir mediation made an efficient excitation energy transfer from Coumarin to Bodipy to trigger the redox reactions. In this process, the Coumarin in **Ir-4** can absorb the light and efficiently convert

excitation energy into Bodipy-localized triplet state by the cascade processes of ISC and intramolecular TTET, which can further promote electron transfer process from **DMT** to excited **Ir-4** and subsequently photocatalytic hydrogen evolution. The BSVLA PS was first introduced into water splitting system, exhibiting significantly enhanced photocatalytic performance >320 times higher than that of typical **Ir-1**, and the TON towards **Ir-4** reaches to 115,840, representing the most active sensitizer among all the molecular systems. Much enhanced photocatalytic activity of **Ir-4** was mainly attributed to its BSVLA ability, long-lived excited state, and delicate synergistic effect between different components. This work paves the way to develop BSVLA multi-component array on molecular level for efficient solar light conversion and boosting artificial photosynthesis.

## Methods

**Materials and methods.** All the reactions were performed in argon unless otherwise mentioned. All the solvents were of analytical grade and distilled before use. $IrCl_3 \cdot 3H_2O$ and Coumarin 6 was purchased from Sigma-Aldrich. 2-Phenylpyridine, $NH_4PF_6$, and N-iodosuccinimide were purchased from HEOWNS. Triphenylphosphine, dichlorobis(triphenyl-phosphine)palladium(II), CuI, and 2,2′-bipyridine were purchased from Adamas-beta. Chromatographic-grade acetonitrile was purchased from Adamas Reagent. The synthetic scheme of **Ir-1**–**Ir-4** is presented in Supplementary Fig. 1. The synthetic intermediates and target complexes were evidenced by $^1H$ NMR and mass spectroscopy.

**Instruments.** The amount of the hydrogen product was analyzed by gas chromatography (Shimadzu GC-2014+AT 230C, TDX-01 column, TCD, argon carrier). UV-vis absorption spectra were recorded on a LAMBDA750 UV-vis spectrophotometer. FL spectra were taken on Hitachi F4600 spectrofluorometer. Transient absorption spectra were measured on the LP980 laser flash photolysis instrument (Edinburgh, UK). FL lifetime was performed by time-resolved confocal FL instrument (MicroTime 200, PicoQuant, Berlin, Germany). Photocatalytic experiments were conducted with 450 and 525 nm LED light (Zolix, MLED4) and 175 W Xenon lamp (LX-175, PEILC, Japan) with 420 nm filter. Electrochemical measurements were carried out on a CHI 760E electrochemical workstation at room temperature.

**Photocatalytic hydrogen production.** Photocatalytic hydrogen evolution was conducted under 1 atm of Ar at 25 °C in 16 mL reactor containing PS (1.25 μM), **C-1** (50 μM), **DMT** (10 mM), 0.5 mL $H_2O$, and 4.5 mL $CH_3CN$. The mixture was continuously stirred and irradiated under a Xe lamp ($\lambda > 420$ nm, 122 mW cm$^{-2}$).

**Spectrum measurement.** All the solvents were chromatographically pure for spectra measurement, and all the measurements were performed under an argon atmosphere unless otherwise stated. For guaranteeing the accuracy of the test outcomes, $Ru(bpy)_3^{2+}$ served as standard sample to correct the instrument and the method of measurement.

**Measurement of apparent quantum efficiency (AQE).** AQE was measured under the same photocatalytic reaction condition, and 175 W Xe lamp fitted with the 475 and 520 nm band-pass filters, and AQE was calculated according to Eq. 1:

$$AQE = (\text{number of reacted electrons/number of incident photons}) \times 100 =$$
$$[(\text{number of evolved } H_2 \text{ molecules} \times 2)/\text{number of incident photons}] \times 100$$

$$(1)$$

The number of incident photons was $2.6 \times 10^{19}$ photons h$^{-1}$ at 475 nm and $2.5 \times 10^{19}$ photons h$^{-1}$ at 520 nm, which was measured by using a calibrated Si photodiode (HAMAMATSU S2281).

**Isotope tracer experiment.** The isotope tracer experiment was performed by using $D_2O$ to replace $H_2O$ in the photocatalytic $H_2$ evolution process while all other constituents remained the same. After the photocatalytic reaction, the gaseous product was analyzed with MS (HIDEN ANALYTICAL, HAS-301–144FPL). The background of $H_2$ was collected from the container containing pure $CH_3CN$ under 1 atm of Ar.

**Spectroelectrochemistry.** SEC experiments were executed in the anhydrous $CH_3CN$ under nitrogen atmosphere. These measurements were performed using controlled-potential electrolysis (CHI 760E electrochemical workstation) at potentials 50–100 mV more negative than the reduction potential or 50–100 mV more positive than the oxidation potential of complexes **Ir-1**–**Ir-4**. Electrolyses was carried out on a reticulated Pt electrode with three-dimensional meshes (0.8 × 0.5 × 0.1 cm$^3$). The auxiliary electrode was a Pt wire. The solution in 1 mm path-length cell mainly contained 0.1 mM PS, 0.6 mL $CH_3CN$, and 0.1 M [$Bu_4N$]$PF_6$. The change in absorption due to the reduction or oxidation of the PSs was

monitored by the HITACHI optic spectrophotometer (U-3900) for UV−visible experiments equipped with both deuterium ($D_2$) and tungsten iodide (WI) light sources (2J1–1500, 885–1200, HITACHI Optics).

**Femtosecond transient absorption spectra.** The pump beam was generated from a regenerative amplified Ti: sapphire laser system from Coherent (800 nm, 100 fs, 6 mJ per pulse, and 1 kHz repetition rate). Output pulse of 800 nm from the regenerative amplifier was split into two parts with a beam splitter. The reflected part was used to pump a TOPAS Optical Parametric Amplifier, which generates a wavelength-tunable laser pulse from 250 nm to 2.5 μm as pump beam. The transmitted 800-nm beam was attenuated with a neutral density filter and focused into a rotating $CaF_2$ disk to generate a white light continuum from 350 to 800 nm used for probe beam. The probe beam was focused with an Al parabolic reflector onto the sample. After penetrating the sample, the probe beam was collimated to focus into a fiber-coupled spectrometer and detected at a frequency of 1 KHz. The intensity of the pump pulse used in this experiment was controlled by a variable neutral-density filter wheel. The delay between the pump and probe pulses was controlled by a motorized delay stage. The pump pulses were chopped by a synchronized chopper at 500 Hz. The pump pulse was kept in a weak regime where the excitonic annihilation effect can be neglected.

**DFT calculation.** The geometries and the spin density surfaces of the complexes (**Ir-1**–**Ir-4**) were performed at the B3LYP/6–31G/LanL2DZ level. There are no imaginary frequencies for all optimized structures of **Ir-1**–**Ir-4**. The UV-vis absorption, the FL, and the triplet state energy levels were carried out with the TDDFT method. VR stands for vibrational relaxation, indicating the transition from $v = x$ ($x > 0$) level of excited state to $v = 0$ level of excited state (Fig. 8). All these calculations were performed with Gaussian 09 program[60].

**Other methods.** Other information about syntheses, characterizations, and cyclic voltammograms of compounds are given in Supplementary Information.

## Data availability
All relevant data underlying Figs. 1b–e, 2, 3a–e, 4–6 and Supplementary Figures are available from the authors.

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

## Acknowledgements

This work was supported by National Key R&D Program of China (2017YFA0700104), the National Natural Science Foundation of China (Nos. 21703155, 21722104, 21671032, 21401095), Natural Science Foundation of Tianjin City of China (18JCQNJC76500/18JCJQJC47700/17JCQNJC05100), and 111 project of China (No. D17003). We are very grateful to Professor Xiyou Li and Dr. Heyuan Liu from China University of Petroleum (East China) and Dr. Jiani Ma from Northwest University (China) for the measurement and analysis of femtosecond transient absorption spectroscopy.

## Author contributions

S.G. and Z.-M.Z. conceived and designed this project, P.W., S.G., K.-K.C. and N.Z. performed the experiments, S.G., H.-J.W., and T.-B.L. carried out the DFT calculation, P.W., S.G., Z.-M.Z., and T.-B.L. analyzed the data, P.W., S.G., Z.M.Z., and T.B.L. wrote and revised the article. All authors participated in drafting the paper and gave approval to the final version of the manuscript.

## Additional information

**Competing interests:** The authors declare no competing interests.

**Peer Review Information:** *Nature Communications* thanks the anonymous reviewer(s) for their contribution to the peer review of this work. Peer reviewer reports are available.

