## [Peer Review File · Nature Communications]

Reviewers' comments:

Reviewer #1 (Remarks to the Author):

Review of Nguyen et al.

This is an interesting new result regarding the intermolecular dynamics of an important benchmark carbon dioxide reduction catalyst. The comparison between solution and solid state provides the kind of information that can distinguish between interactions with the TEOA sacrificial donor and the complex from those that are intramolecular to the complex itself.

The terahertz technique is somewhat limited with regard to the type of interactions it can probe and the resolution of the features in the spectra require some care to be useful. The authors have provided an interpretation of these spectra features that is plausible, yet it would be useful to compare this rather specialized data with other spectral techniques that might help solidify the interpretation. I know that this might be difficult, yet the discussion seems quite speculative given the nature of the terahertz data. Nevertheless, this work presents new results using a technique that may be more generally useful in understanding more subtle vibrational interactions in other photocatalysts as well, and should be published.

Reviewer #2 (Remarks to the Author):

In their article entitled "A broadband and strong visible-light-absorbing photosensitizer boosts hydrogen evolution" P. Wang and al. report on the use of a novel cationic Ir complex, bearing coumarin-6 and Bodipy antenna as photosensitizer (PS) for hydrogen evolution. A significant increase of the photosensitizing performances was observed for the latter complex when compared to references complexes decorated with none or only one of the two types of organic chromophores, when used with a standard catalyst (cobaloxime) in mixed organic-aqueous media. The authors attribute this improved activity to the nature of the excited state of these complexes (long-lived triplet) and efficient energy transfer between the organic chromophores allowing for an optimized use of the available incident light.

Despite proposing an interesting set of data and one example of the successful use of intramolecular antenna effect applied to photo-catalysis, the manuscript lacks the expected quality for publication in Nature Communications in its current version. Therefore I suggest the author to revise their manuscript before re-evaluation.

Regarding the photocatalysis experiments:

a) One of the main selling point of this study is the report of the highest turnover number observed for any known PS so far, for Ir4. The conditions in which this result was obtained are very specific. The concentration of the PS (Ir4) is set to as low as 12.5 nM when the cobaloxime concentration was set to 0.1 mM and the sacrificial electron donor (DMT) to 60 mM. For a fair comparison, the authors should test and discuss the TON of all the dyes studied, under the same optimized conditions.

b) The actual source of protons in these systems is not adequately discussed. The authors should consider protons released from the oxidized sacrificial electron donor (J. Electroanal. Chem. 2002, 531, 33). The direct reduction of protons from water appears highly unlikely as shown by the absence of any catalytic wave associated with the CoII/CoI couple in ACN/H₂O 9/1 (Figure S24 b).

c) The catalytic cycle proposed in Scheme 2 and discussed in the text does not account for the well accepted mechanism of proton reduction by cobaloximes, which implies Co(III)-H hydride intermediate and reduction of the latter in the absence of a strong acid (Inorg. Chem. 2013, 52, 6994).

d) The stability of the different components of the system is not discussed. Why does the system stop evolving hydrogen after a few hours of irradiation? Can the activity be restored by addition of PS, catalyst, or DMT?

Regarding the physico-chemical analysis:

a) I note that Ir1, Ir2, and Ir3 complexes are known complexes with well-described physico-chemical properties. One can expect the authors to refer explicitly to the existing literature, when reproducing some of the previously published work and analysis, and discuss their own results in light of the previously published studies. Particularly for Ir2 (Eur. J. Inorg. Chem., 2012, 3975; Inorg. Chem., 2016, 55, 8723) and Ir3 (Inorg. Chem., 2013, 52, 6299).

b) The discussion regarding the excited state dynamics of Ir4 appears confusing. The authors mention "...DFT/TDDFT confirming the efficient electron transfer from coumarin to Bodipy and the population of Bodipy-localized 3IL state..." while in the rest of the text they propose a fast and efficient triplet-triplet energy transfer between the coumarin and Bodipy fragments. Please clarify.

c) The "conformational transformation" (CT) introduced in Figure 7 is never discussed in the main text.

Miscellaneous remarks:

a) The authors must explicitly cite the references on which they base their work when reproducing previously published synthesis (L2, Ir1, Ir2, Ir3) even with minor changes in the procedures.

b) Please find here are a few minor points that require attention before publication:

- "DMT" must be defined before its first use in the main text

- the excitation wavelength (525 nm) indicated on Figure 1c (caption) does not correspond to the excitation wavelength given in the main text (532 nm)

- please correct the excitation wavelength indicated on Figure S19 c (caption)

- figure S26 e and S26 f are missing.

Reviewer #3 (Remarks to the Author):

The article presents a new multichromophoric complex for homogenous photocatalytic water reduction. The complex (Ir-4 in Scheme 1) was derived from the traditional catalysts in Ir(ppy)₂(bpy)⁺ (Ir-1 in Scheme 1) by replacing the ppy ligands with coumarin 6 and bodipy derivatives. Compared to Ir-1 as the chromophore, the new complex absorbs broadly in the visible spectrum with enhanced absorptivities. In a photocatalytic system with a sacrificial electron donor (DMT) and a H₂ evolution catalyst (C-1), the excited state(s) of Ir-4 can initiate a sequence of efficient electron transfer reactions towards water reduction. The integrated photosystem is active for water reduction for ~18 h with a large TON. The authors use nanosecond transient absorption, steady-state emission, cyclic voltammetry and DFT calculations to support the conclusions that the enhanced photocatalytic efficiency of the system originates from a combination of energetically favored charge transfer between the functional groups (coumarin 6, bodipy and Ir-bpy) in Ir-4 and between Ir-4 and added DMT. The idea on the design of multichromophoric complex is of interest. However, some of the

conclusions cannot be supported by the data and analysis. Revisions are suggested below.

1. There is no direct evidence that the excited state of Ir-4 reacts first with DMT in a system in the presence of both DMT and C-1. In fact, the results in Table 1 show much a larger K_s - v value for the quenching by C-1 than by DMT. It is possible that the excited state is first oxidatively quenched by C-1 and that the oxidized form of oxidized Ir-4 reacts with DMT. Can the authors exclude that possibility?
2. In order to analyze the transient absorption data in Figures 4, 5, S26 and S27, spectroelectrochemical data showing the spectral features of the reduced and oxidized forms are necessary. Without the data, there is no evidence to support the assignment of the observed transient absorptive features and the conclusions on the origin of formation and decay.
3. When discussing the transient absorption figures on pages 11-14, please refer to specific figure panels in the text for different photosystems.
4. In line 153 on page 8, the authors mention that the PL of Ir-3 and Ir-4 are weak or absent. Can the authors explain the cause of the phenomenon? Based on the cause, how do the authors evaluate Stern-Volmer plots without the observation of PL?
5. The driving force for the second electron reduction of C-1 is negligible (~ 0.02 eV). Is there any evidence for that reaction?
6. In Scheme 2, energy transfer is not proven by presentation of data or analysis.
7. The irradiation source for the photocatalysis has an intense feature at 175 W. Please add the quantum yield based on the incident photons for evaluation of the photocatalytic efficiency.
8. Please add a plot of the TOF as a function of irradiation time for the Ir-4 containing catalyst to show the change of H₂ evolution rate with time.
9. Please provide mass spectra for the new complexes Ir-2, Ir-3 and Ir-4.

Response to Reviewers' comments

To Reviewer 2

In their article entitled "A broadband and strong visible-light-absorbing photosensitizer boosts hydrogen evolution" P. Wang and al. report on the use of a novel cationic Ir complex, bearing coumarin-6 and Bodipy antenna as photosensitizer (PS) for hydrogen evolution. A significant increase of the photosensitizing performances was observed for the latter complex when compared to references complexes decorated with none or only one of the two types of organic chromophores, when used with a standard catalyst (cobaloxime) in mixed organic-aqueous media. The authors attribute this improved activity to the nature of the excited state of these complexes (long-lived triplet) and efficient energy transfer between the organic chromophores allowing for an optimized use of the available incident light.

Despite proposing an interesting set of data and one example of the successful use of intramolecular antenna effect applied to photo-catalysis, the manuscript lacks the expected quality for publication in Nature Communications in its current version. Therefore I suggest the author to revise their manuscript before re-evaluation.

Reply: Thanks very much for your kind comments and valuable suggestions, we have tried our best to address these issues in this work.

Question 1. Regarding the photocatalysis experiments:

(a) One of the main selling point of this study is the report of the highest turnover number observed for any known PS so far, for **Ir-4**. The conditions in which this result was obtained are very specific. The concentration of the PS (**Ir-4**) is set to as low as 12.5 nM when the cobaloxime concentration was set to 0.1 mM and the sacrificial electron donor (**DMT**) to 60 mM. For a fair comparison, the authors should test and discuss the TON of all the dyes studied, under the same optimized conditions.

Reply: Thanks a lot for your valuable suggestion. Accordingly, we have investigated the photocatalytic activity of **Ir-1**, **Ir-2** and **Ir-3** under the same optimized conditions as those of **Ir-4** for a fair comparison. It can be found that the TONs of **Ir-1**, **Ir-2**, **Ir-3**

and **Ir-4** are 361, 22560, 8270 and 115840, respectively (Figure S17), in which the TON of **Ir-4** significantly outperforms those of other PSs, and is over 320 times higher than that of **Ir-1**, demonstrating **Ir-4** is indeed a state-of-the-art PS so far. The comments and experimental results were supplied in the revised manuscript (Page 5, line 100) and supporting information (SI) (Figures S16 and S17), respectively.

(b) The actual source of protons in these systems is not adequately discussed. The authors should consider protons released from the oxidized sacrificial electron donor (J. Electroanal. Chem. 2002, 531, 33). The direct reduction of protons from water appears highly unlikely as shown by the absence of any catalytic wave associated with the $\text{Co}^{\text{II}}/\text{Co}^{\text{I}}$ couple in ACN/ H_2O 9/1 (Figure S24b).

Reply: Thanks for your kind suggestion. In order to ascertain the actual source of protons, photocatalytic experiments with H_2O and D_2O were performed, and the products were determined by mass spectrometry (MS) analysis. As shown in Figure S19, H_2 as the sole gaseous product was detected by MS in the mixed solvents of H_2O and CH_3CN , and D_2 became the major product by replacing H_2O with D_2O in the photocatalytic system, indicating that H_2 does derive from the H_2O in this photocatalytic system. In addition, the photocatalytic experiment with pure CH_3CN as solvent was also performed (Table S2, in SI). The result showed that only trace amount of hydrogen can be detected in the absence of H_2O , further excluding the dehydrogenation of **DMT** or CH_3CN to release H_2 . Related description was discussed in the revised manuscript (Page 7, line 141) and experimental results were supplied in SI (Table S2 and Figure S19).

I agree with your opinion that proton reduction was almost impossible to happen at the position of $\text{Co}(\text{II})/\text{Co}(\text{I})$ couple due to its reversible redox feature. In order to observe the catalytic wave, CV of **C-1** was scanned to the more negative position and a strong catalytic current was observed at around -1.48 V (vs. SCE). According to the previously reported results, this potential could be tentatively attributed to $\text{Co}(\text{I})/\text{Co}(\text{0})$ couple (Acc. Chem. Res., 2009, 42, 1995-2004; Dalton Trans., 2015, 44,

17704-17711). The comments and experimental results were supplied in the revised manuscript (Page 11, line 223) and SI (Figure S30).

(c) The catalytic cycle proposed in Scheme 2 and discussed in the text does not account for the well accepted mechanism of proton reduction by cobaloximes, which implies Co(III)-H hydride intermediate and reduction of the latter in the absence of a strong acid (Inorg. Chem. 2013, 52, 6994).

Reply: In this work, we have devoted most of our attention to the synthesis of broadband and strong visible-light-absorbing photosensitizer, and just employed the standard catalyst (cobaloxime) to evaluate their performance. According to your kind suggestion, we tried our best to explain the catalytic mechanism in this revised version. As shown in Figure S30, the redox process of Co(II)/Co(I) remained reversible, indicating that the mechanism associated with Co(III)-H hydride intermediate is improper. Interestingly, a strong catalytic current formed at around -1.48 V (*vs.* SCE), which could be attributed to the proton reduction process in the presence of Co(0) species (Acc. Chem. Res., 2009, 42, 1995-2004; Dalton Trans., 2015, 44, 17704-17711). Considering that the possibility of electron transfer from reduced **Ir-4** to Co(I) was thermodynamically ruled out, the source of Co(0) species could be tentatively attributed to the disproportionation of Co(I) into Co(II) and Co(0). Further, electrolysis experiment of **C-1**-containing system was performed to confirm this view, where H₂ was indeed detected at -1.09 V (*vs.* SCE, corresponding to Co(II)/Co(I)), indicating the formation of Co(0) species during this electrolysis process. In addition, the disproportionation of Co(I) into Co(II) and Co(0) has been confirmed by the previously published work (Science, 2018, 360, 888-893). As a result, the catalytic process in scheme 2 was revised in the revised manuscript. The comments were added to the revised manuscript (Page 11, line 223 and Page 18, line 388) and SI (Figure S30 and Table S4).

(d) The stability of the different components of the system is not discussed. Why does the system stop evolving hydrogen after a few hours of irradiation? Can the activity

be restored by addition of PS, catalyst, or **DMT**?

Reply: Thanks a lot for this valuable suggestion. In order to investigate the reason of catalytic deactivation after a few hours of irradiation, we performed the recycle photocatalytic experiments by re-adding PS, catalyst, or **DMT**. It could be observed that photocatalytic activity of **Ir-4**-containing system partially restore when the **DMT**, **C-1** or **DMT / C-1** was added into the reaction system. In contrast, minor amount of H₂ can be detected by re-addition of **Ir-4**, indicating that the inactivation of photocatalytic system was mainly due to the decomposition of the **DMT** and **C-1**. The comments (Page 6, line 111) and experimental results (Figure S18) were supplied in the manuscript and SI, respectively.

Question 2. Regarding the physico-chemical analysis:

a) I note that **Ir-1**, **Ir-2**, and **Ir-3** complexes are known complexes with well-described physico-chemical properties. One can expect the authors to refer explicitly to the existing literature, when reproducing some of the previously published work and analysis, and discuss their own results in light of the previously published studies. Particularly for **Ir-2** (Eur. J. Inorg. Chem., 2012, 3975; Inorg. Chem., 2016, 55, 8723) and **Ir-3** (Inorg. Chem., 2013, 52, 6299).

Reply: According to your kind suggestion, the complexes **Ir-1**, **Ir-2**, and **Ir-3** were explicitly cited, particularly for **Ir-2** and **Ir-3**. The detail citations were added to the revised manuscript (in pages 3, 4, 7, 8, 9 and 13). Thanks a lot.

b) The discussion regarding the excited state dynamics of **Ir-4** appears confusing. The authors mention "...DFT/TDDFT confirming the efficient electron transfer from coumarin to Bodipy and the population of Bodipy-localized ³IL state..." while in the rest of the text they propose a fast and efficient triplet-triplet energy transfer between the coumarin and Bodipy fragments. Please clarify.

Reply: Thanks for your kind reminding. We have updated this sentence as "... DFT/TDDFT confirming the efficient triplet energy transfer from coumarin to Bodipy

and the population of Bodipy-localized ^3IL state...” in the revised manuscript. (in page 17, line 366)

Further, we performed the femtosecond transient absorption of complexes **Ir-2-Ir-4** to clarify the energy transfer process. A fast intramolecular triplet energy transfer from Coumarin 6 to Bodipy was confirmed and its rate constant was determined as $k_{\text{TET}} = 4.3 \times 10^{10} \text{ s}^{-1}$. The comments (in page 15, line 315) and spectra (Figure S36) were supplied in the revised manuscript and SI, respectively.

c) The “conformational transformation” (CT) introduced in Figure 7 is never discussed in the main text.

Reply: Thanks for your kind suggestion. The “conformational transformation” (CT) has been replaced by “vibrational relaxation” (VR) in the main text (in Page 17, Figure 7). VR stands for the transition from $v = x$ ($x > 0$) level of excited state (e.g. S_2) to $v = 0$ level of excited state (S_2) accompanied by thermal radiation in the solution. (Turro, N. J. et al. Principles of Molecular Photochemistry: An Introduction, 1st edition.; University Science Books: California, 2008; Smith, K. C. Courseware of Basic Photochemistry, 2014)

Question 3. Miscellaneous remarks:

a) The authors must explicitly cite the references on which they base their work when reproducing previously published synthesis (**L-2, Ir-1, Ir-2, Ir-3**) even with minor changes in the procedures.

Reply: According to your kind suggestions, we have explicitly cited the references when reproducing previously published synthesis (**L-2, Ir-1, Ir-2, Ir-3**). The citations were labeled in the revised SI (Pages S6-S7).

b) Please find here are a few minor points that require attention before publication:

- “**DMT**” must be defined before its first use in the main text
- the excitation wavelength (525 nm) indicated on Figure 1c (caption) does not correspond to the excitation wavelength given in the main text (532 nm)
- please correct the excitation wavelength indicated on Figure S19 c (caption)

- figure S26 e and S26 f are missing.

Reply: Thanks a lot for your kind suggestions.

- **DMT** has been defined as “N, N-dimethyl-p-toluidine” in the caption of Table 1 in the revised manuscript in Page 5.
- the excitation wavelength given in the main text (532 nm) has been change to 525 nm.
- the excitation wavelength has been updated as 450 nm in the revised SI, in Figure S19c (update to Figure S23c).
- The caption of Figure S26e and S26f (update to Figure S31e and S31f) were supplied in the revised supporting information.

To Reviewer 3

The article presents a new multichromophoric complex for homogenous photocatalytic water reduction. The complex (**Ir-4** in Scheme 1) was derived from the traditional catalysts in $\text{Ir}(\text{ppy})_2(\text{bpy})^+$ (**Ir-1** in Scheme 1) by replacing the ppy ligands with coumarin 6 and bodipy derivatives. Compared to **Ir-1** as the chromophore, the new complex absorbs broadly in the visible spectrum with enhanced absorptivities. In a photocatalytic system with a sacrificial electron donor (**DMT**) and a H_2 evolution catalyst (**C-1**), the excited state(s) of **Ir-4** can initiate a sequence of efficient electron transfer reactions towards water reduction. The integrated photosystem is active for water reduction for ~18 h with a large TON. The authors use nanosecond transient absorption, steady-state emission, cyclic voltammetry and DFT calculations to support the conclusions that the enhanced photocatalytic efficiency of the system originates from a combination of energetically favored charge transfer between the functional groups (coumarin 6, bodipy and Ir-bpy) in **Ir-4** and between **Ir-4** and added **DMT**. The idea on the design of multichromophoric complex is of interest. However, some of the conclusions cannot be supported by the data and analysis. Revisions are suggested below.

Reply: Thanks a lot for your professional suggestions. Accordingly, we performed a series of additional experiments to address the issues you concerned. The detail responses are listed as following.

Question 1. There is no direct evidence that the excited state of **Ir-4** reacts first with **DMT** in a system in the presence of both **DMT** and **C-1**. In fact, the results in Table 1 show much a larger K_{s-v} value for the quenching by **C-1** than by **DMT**. It is possible that the excited state is first oxidatively quenched by **C-1** and that the oxidized form of oxidized **Ir-4** reacts with **DMT**. Can the authors exclude that possibility?

Reply: Thanks a lot for your kind suggestion. The quenching constants of **Ir-4** by **DMT** and **C-1** in Table 1 were obtained from the fluorescence quenching experiments, and their K_{s-v} values were no more than 2000 M^{-1} , indicating an inefficient electron transfer between singlet excited state **Ir-4** and **DMT** (or **C-1**). This result could be attributed to its short fluorescence lifetime (0.14 ns at 560 nm) (J. Am. Chem. Soc. 2009, 131, 9192–9194; J. Am. Chem. Soc., 2010, 132, 15480–15483). As a result, the K_{s-v} value of **Ir-4** by **C-1** or **DMT** in Table 1 cannot reflect the actual electron transfer efficiency in the photocatalytic process. According to the results of nanosecond transient absorption, electron transfer process between different components should be dominated by long-lived triplet state of **Ir-4**.

As well known, photocatalytic hydrogen evolution could proceed by two different photochemical routes: an oxidative mechanism and a reductive mechanism. In this work, the photocatalytic process of **Ir-4**-containing system was proposed as reductive mechanism based on the following three main reasons: i) Nanosecond transient absorption spectrum of **Ir-4** with **DMT** and **C-1** showed the formation of reduced **Ir-4**, and the lifetime of reduced **Ir-4** became shorter with increasing the concentration of **C-1**. These results indicate that the excited **Ir-4** accepted electrons firstly from **DMT** to generate the reduced **Ir-4**, which can deliver electrons to catalyst subsequently; ii) The Gibbs free energy changes (ΔG_{CS}) of electron transfer from excited **Ir-4** to **C-1** (Co^{2+}) was determined to be + 0.31, indicating that the oxidative mechanism by **C-1** can be thermodynamically ruled out; iii) the concentration of **DMT** is much higher

than that of **C-1** under the catalytic condition, further supporting the reduction mechanism (ChemPlusChem. 81, 1090–1097). Based on the above results, the possibility that the excited state of **Ir-4** is first oxidatively quenched by **C-1** can be excluded.

Question 2. In order to analyze the transient absorption data in Figures 4, 5, S26 and S27, spectroelectrochemical data showing the spectral features of the reduced and oxidized forms are necessary. Without the data, there is no evidence to support the assignment of the observed transient absorptive features and the conclusions on the origin of formation and decay.

Reply: According to your kind suggestion, spectroelectrochemical data of complexes **Ir-1-Ir-4** were supplied in the revised supporting information (Figures S34- S35). The transient absorption spectra of **Ir-1-Ir-4** with **DMT** well match with the absorption of reduced PSs, but were different from that of the oxidized ones. For example, transient absorption spectrum of **Ir-3** with **DMT** shows two positive absorption band at around 440 nm and 580 nm, which well matches with the differential spectrum of UV-vis absorption of reduced **Ir-3** (Figure R1). These results confirm that the transient absorption spectra of **Ir-1-Ir-4** with **DMT** could be assigned to the absorption of reduced PSs. The detail comments were added to the revised manuscript (in page 15, line 310) and the spectra were supplied in the revised supporting information (Figures S34-S35).

Figure R1. Nanosecond transient absorption spectra of (a) **Ir-3** in the presence of 40 mM of **DMT**. The differential spectra of UV-vis absorption of (b) reduced **Ir-3**, (c)

oxidized **Ir-3**. These spectra in (a) and (b) were recorded in situ with a spectroelectrochemical cuvette containing 1.0×10^{-4} M PSs in deaerated CH₃CN.

Question 3. When discussing the transient absorption figures on pages 11-14, please refer to specific figure panels in the text for different photosystems.

Reply: Thanks for your kind suggestion. We have referred to specific Figure panels in the text for different photosystems, as discussing the transient absorption Figures on pages 12-15.

Question 4. In line 153 on page 8, the authors mention that the PL of **Ir-3** and **Ir-4** are weak or absent. Can the authors explain the cause of the phenomenon? Based on the cause, how do the authors evaluate Stern-Volmer plots without the observation of PL?

Reply: The PL of **Ir-3** and **Ir-4** are weak or absent, which could be attributed to the following reasons: i) Photophysical radiationless deactivations in solution at room temperature, such as bimolecular and diffusional quenching processes; ii) The transition of T→S₀ is forbidden in general. This situation is generally of low probability to emit phosphorescence. The exception is heavy atom effect, which can enhance spin-orbit coupling and further promote the probability of the radiative transition of T→S₀. The heavy atom effect of **Ir-3** and **Ir-4** is much weaker than that of **Ir-1** and **Ir-2** due to the relative far distance between Ir atom and the center of ligand, indicating a weak spin-orbit coupling for **Ir-3** and **Ir-4**. Thus this result will be harmful to their phosphorescence emission. (Turro, N. J. et al. Principles of Molecular Photochemistry: An Introduction, 1st edition.; University Science Books: California, 2008) iii) Both **Ir-3** and **Ir-4** have a long-lived excited states (101 μs for **Ir-3** and 89 μs for **Ir-4**), which can provide more possibilities to encounter with quenchers.

Fortunately, although the PL of **Ir-3** is weak, we can get its stern-volmer quenching constant by quenching experiment with **C-1** and **DMT** (see Table 1 in the revised manuscript). However, we cannot get the actual stern-volmer quenching constant of **Ir-4** due to no PL for **Ir-4**.

Question 5. The driving force for the second electron reduction of **C-1** is negligible (~0.02 eV). Is there any evidence for that reaction?

Reply: The photocatalytic processes of **Ir-3** and **Ir-4**-containing system have been determined as reduction mechanism. In this reduction mechanism, **DMT** firstly donates electrons to the excited PS to yield the reduced PS, which can successively transfer two electrons to Co^{3+} to generate Co^+ , and the ΔG_{CS} for above processes are all negative, indicating the thermodynamic feasible for relevant electron transfer. As mentioned above, the driving force for the second electron reduction of **C-1** is only 0.02 for **Ir-3** and **Ir-4**-containing system, which is much smaller than **Ir-1**. However, the photocatalytic activity of **Ir-4** was over 21 times higher than that of **Ir-1**. As a result, we proposed that broadband and strong visible-light-absorbing ability and long-lived excited state of **Ir-4** can redeem its humble driven force for electron transfer. According to your kind suggestion, we tried our best to explain the catalytic mechanism in this revised version, which was supplied in the main text (in page 12, line 246).

Question 6. In Scheme 2, energy transfer is not proven by presentation of data or analysis.

Reply: Thanks a lot for your kind suggestions. In order to prove the intramolecular energy transfer process of **Ir-4**, femtosecond transient absorption spectroscopy was performed. As a result, a fast intramolecular triplet energy transfer from Coumarin 6 to Bodipy occurs and its rate constant is determined as $k_{\text{TET}} = 4.3 \times 10^{10} \text{ s}^{-1}$. Further, DFT calculation rationalizes the photophysical processes of **Ir-4**. The detail comments were added to the revised manuscript (in page 15, line 315) and their spectra were supplied in the revised supporting information (Figure S36).

Question 7. The irradiation source for the photocatalysis has an intense feature at 175 W. Please add the quantum yield based on the incident photons for evaluation of the photocatalytic efficiency.

Reply: According to your kind suggestion, we have determined the quantum yield with the values of 37.7 % at 475 nm and 25.1 % at 520 nm based on the incident photons, which have been supplied in the revised manuscript (in page 7, line 140). The detail measurement method has been supplied in the SI in page S2.

Question 8. Please add a plot of the TOF as a function of irradiation time for the **Ir-4** containing catalyst to show the change of H₂ evolution rate with time.

Reply: According to your kind suggestion, a plot of the TOF as a function of irradiation time for **Ir-4** was added to the revised SI (Figure S15). As shown in Figure S15, H₂ evolution rate gradually decreased with increasing of the reaction time. This result can be mainly attributed to the decomposition of **DMT** and **C-1** in the photocatalytic process (see response to **question 1d** to **reviewer 2**).

Question 9. Please provide mass spectra for the new complexes **Ir-2**, **Ir-3** and **Ir-4**.

Reply: According to your kind suggestion, the mass spectra for complexes **Ir-2**, **Ir-3** and **Ir-4** have been provided in the revised manuscript (Figure S9, Figure S11 and Figure S13), which further confirm the successful synthesis of these PSs.

Thanks a lot!

REVIEWERS' COMMENTS:

Reviewer #2 (Remarks to the Author):

In the revised manuscript, the authors have answered the main questions that remained in the original version. I therefore recommend publication of this article in Nature Communications.

Reviewer #3 (Remarks to the Author):

The authors' responses and the corresponding revisions in the manuscript are satisfactory. I recommend it for publication in its current form.

Response to Reviewers' comments

To Reviewers

Reviewer #2 (Remarks to the Author):

In the revised manuscript, the authors have answered the main questions that remained in the original version. I therefore recommend publication of this article in Nature Communications.

Reviewer #3 (Remarks to the Author):

The authors' responses and the corresponding revisions in the manuscript are satisfactory. I recommend it for publication in its current form.

Response: Thanks a lot for your hard work and nice comments in reviewing our manuscript.

Thanks a lot!